# Extraplexus versus intraplexus ultrasound-guided interscalene brachial plexus block for ambulatory arthroscopic shoulder surgery: A randomized controlled trial

Monica W. Harbell[1,2]*, Kerstin Kolodzie[1,3], Matthias Behrends[1], C. Benjamin Ma[4], Sakura Kinjo[1], Edward Yap[1,5], Matthias R. Braehler[1], Pedram Aleshi[1]

1 Department of Anesthesia and Perioperative Care, University of California, San Francisco, California, United States of America, 2 Department of Anesthesia and Perioperative Medicine, Mayo Clinic, Phoenix, Arizona, United States of America, 3 Department of Epidemiology and Biostatistics, University of California, San Francisco, California, United States of America, 4 Department of Orthopedic Surgery, University of California, San Francisco, California, United States of America, 5 Department of Anesthesia, Kaiser Permanente South San Francisco, South San Francisco, California, United States of America

* Monica.Harbell@ucsf.edu

## Abstract

### Background

This randomized study compared the efficacy and safety of extraplexus and intraplexus injection of local anesthetic for interscalene brachial plexus block.

### Methods

208 ASA I-II patients scheduled for elective shoulder arthroscopy under general anesthesia and ultrasound-guided interscalene brachial plexus block were randomly allocated to receive an injection of 25mL ropivacaine 0.5% either between C5-C6 nerve roots (intra-plexus), or anterior and posterior to the brachial plexus into the plane between the perineural sheath and scalene muscles (extraplexus). The primary outcome was time to loss of shoulder abduction. Secondary outcomes included block duration, perioperative opioid consumption, pain scores, block performance time, number of needle passes, onset of sensory blockade, paresthesia, recovery room length of stay, patient satisfaction, incidence of Horner's syndrome, dyspnea, hoarseness, and post-operative nausea and vomiting.

### Results

Time to loss of shoulder abduction was faster in the intraplexus group (log-rank p-value<0.0005; median [interquartile range]: 4 min [2–6] vs. 6 min [4–10]; p-value <0.0005). Although the intraplexus group required fewer needle passes (2 vs. 3, p<0.0005), it resulted in more transient paresthesia (35.9% vs. 14.5%, p = 0.0004) with no difference in any other secondary outcome.

**Data Availability Statement:** All relevant data are within the manuscript and its Supporting Information files.

**Funding:** this study was supported by internal funding by the University of California, San Francisco Department of Anesthesia and Perioperative Care.

**Competing interests:** The authors have declared that no competing interests exist.

## Conclusion

The intraplexus approach to the interscalene brachial plexus block results in a faster onset of motor block, as well as sensory block. Both intraplexus and extraplexus approaches to interscalene brachial plexus block provide effective analgesia. Given the increased incidence of paresthesia with an intraplexus approach, an extraplexus approach to interscalene brachial plexus block is likely a more appropriate choice.

## Introduction

Ultrasound-guided interscalene brachial plexus blocks provide effective analgesia after arthroscopic shoulder surgery and have been shown to reduce opioid consumption, decrease postoperative nausea and vomiting (PONV), improve patient satisfaction, and decrease recovery room length of stay [1–3]. There is interest in identifying the optimal location for local anesthetic administration during an interscalene block. The injection site chosen for any type of block can affect its quality, time to onset, and duration. For example, the injection of local anesthesia into the paraneural sheath during a popliteal sciatic nerve block can result in a faster onset and longer duration of the block [4]. Franco and colleagues identified a sheath surrounding the brachial plexus in a cadaveric study [5]. This has generated interest in comparing intraplexus injection (between C5 and C6 nerve roots within the sheath surrounding the brachial plexus) to extraplexus injections (outside of the brachial plexus sheath) for interscalene brachial plexus block. Previous studies that compared extraplexus to intraplexus injections have had conflicting results with one reporting no difference in block onset [6] and the other reporting a faster onset with the intraplexus approach, but increased paresthesia [7].

We performed a prospective, randomized study to investigate the efficacy and safety of an intraplexus injection between C5 and C6 nerve roots and an extraplexus injection anterior and posterior to the brachial plexus into the plane between the perineural sheath and scalene muscles. This study was registered at Clinicaltrials.gov, identifier: NCT01877330, submitted December 2012, but not publicly posted until June 2013.

## Methods

After study approval from the Institutional Review Board at the University of California, San Francisco on January 16, 2013 (Study #12–10146), consent was informed and written informed consent was obtained from adult patients who were scheduled for arthroscopic shoulder surgery at an ambulatory surgery center between March 2013 and February 2016. All patients who were age ≥ 18 years, American Society of Anesthesiologists (ASA) Physical Status I to II classification, and scheduled for arthroscopic shoulder surgery by one of two surgeons were identified for possible inclusion in the study. Exclusion criteria included: inability to consent, non-English speaking, planned open shoulder surgery, any contraindication for regional anesthesia, such as allergy to local anesthetics, coagulopathy or severe thrombocytopenia, infection at puncture site, pre-existing neuropathy in operative limb, inability to abduct the shoulder to 90 degrees preoperatively, need for postoperative nerve function monitoring, pulmonary disease, low baseline oxygen saturation, dementia, patient refusal, and high preoperative opioid requirements.

Patients were randomly assigned to receive either an intraplexus or an extraplexus injection by computer-generated simple randomization. The allocation sequence was determined prior

to patient enrollment and was concealed to all investigators and anesthesiologists involved in the trial. The random group assignment was stored in consecutively numbered sealed opaque envelopes which were only opened by the anesthesiologist performing the block after informed consent was signed. While the anesthesiologist performing the block was not blinded to group allocation once enrolled and randomized, the operating room team, post-anesthesia care team and research staff recording the sensory and motor data were blinded to group assignment.

## Procedures

The interscalene brachial plexus blocks were performed preoperatively with the patient in the supine position with the head of bed elevated 30 degrees and the patient's head slightly turned away from the operative side. For the block placement, patients received midazolam 0-2mg and/or fentanyl 0–100 mcg intravenously, according to the anesthesiologist's discretion. Using sterile technique, a 6-13-MHz linear probe (LOGIQe, GE Healthcare, Wauwatosa, WI, USA) transducer was used to identify the C5-C6-C7 nerve roots of the brachial plexus. 2–3 mL of lidocaine 2% was used to anesthetize the skin using a 27-G needle. A 5 cm, 22-gauge insulated needle (SonoPlex Stim cannula, Pajunk, Geisingen, Germany) was used to perform a real-time ultrasound-guided interscalene nerve block using a posterior-to-anterior, in-plane needle insertion approach. A total of 25mL of ropivacaine 0.5% was injected in either the intraplexus or extraplexus location (S1 File). In the intraplexus group, the needle was advanced in a posterior-to-anterior direction through the middle scalene muscle, through the brachial plexus sheath in-between the C5-C6 nerve roots. Less than 1 mL of the local anesthetic was injected to confirm that the needle was positioned in the intraplexus location. The remainder of the local anesthetic was injected in divided doses. If the injection did not result in the expansion of the brachial plexus sheath, then it was repositioned so that the tip was within the brachial plexus sheath. In the extraplexus group, the needle was advanced above the C5 nerve root and anterior to the brachial plexus. One mL of local anesthetic was given to confirm the extraplexus location anterior to the brachial plexus sheath. A total of 12mL of ropivacaine 0.5% was administered in anterior to the brachial plexus. The needle was then withdrawn to the posterior surface of the brachial plexus and an additional 12mL of ropivacaine 0.5% was given. If any swelling of the brachial plexus sheath was seen, the needle was repositioned to the extraplexus location. All patients received less than 3 mg/kg of ropivacaine. Patients were instructed to notify the study team if they experienced any paresthesia during block placement. Paresthesia was defined as an abnormal sensation either during or after block placement that felt like "tingling, pins and needles, or electricity down the arm." Anesthesia providers were instructed to stop the block procedure if a paresthesia occurred, to redirect the needle, and to not resume local anesthetic administration until the paresthesia resolved. The block performance time, number of needle passes, and paresthesias were recorded by the study team.

After block placement, sensory and motor examinations were performed at one minute intervals, until there was decrease in sensory exam or until the patient left the preoperative area for the operating room. Patients were assessed for time to complete loss of shoulder abduction by having the patient sit upright and attempt to abduct the arm to 90 degrees. Sensation was assessed using pinprick and comparing to the contralateral side in the sensory distributions of the brachial plexus (axillary, supraclavicular, musculocutaneous, median, ulnar and radial). The time when the sensation first started to feel less sharp compared to the contralateral, non-blocked side was recorded to the nearest minute. Patients were also assessed for signs of Horner's syndrome, hoarseness and dyspnea post-block administration.

All patients then underwent a standardized general anesthetic with a propofol induction and laryngeal mask airway placement. Anesthesia was maintained using a combination of

volatile anesthetics (sevoflurane or desflurane with oxygen/air mixture) and propofol infusion (25–100 mcg/kg/min). Patients received dexamethasone 4 mg IV after induction and ondansetron 4 mg IV prior to emergence as PONV prophylaxis. Fentanyl was administered intravenously during the case per the anesthesiologist's discretion. Sensory exam was reassessed in the recovery room using pinprick. Pain scores were assessed using the Numerical Rating Scale (NRS) from 0–10, where zero is no pain and ten is the worst imaginable pain. Highest NRS pain score in recovery room, perioperative opioid and antiemetic use were recorded, as well as recovery room length of stay and incidence of PONV. Patients were counselled to take oral analgesics on a schedule for the first 24 hours post-operatively.

Patients were contacted by telephone on the day following surgery (at least 24 hours after block placement) to assess block duration, highest pain score at rest and with movement in the 24 hours after block placement, opioid consumption, post-discharge nausea/vomiting and patient satisfaction score. Pain scores were also recorded one week postoperatively.

## Data collection and measurements

Study data were collected and managed using REDCap electronic data capture tools, which is a secure, web-based application designed to support data capture for research studies [8].

**Efficacy outcomes.** The primary outcome of this study was the time to onset of motor blockade, which was defined as the time to complete loss of shoulder abduction. Secondary efficacy outcomes included time to onset of sensory blockade, block performance time, number of needle passes, block duration, block failure, perioperative opioid consumption, pain scores, recovery room length of stay, incidence of PONV and patient satisfaction. These outcomes are further defined in Table 1.

**Table 1. Outcome definitions.**

| | Outcomes | Definition |
|---|---|---|
| **Efficacy** | Motor block onset | Time to complete loss of shoulder abduction (min) |
| | Sensory block onset | Time to altered sensation relative to contralateral side in different peripheral nerve distributions (min) |
| | Block performance time | Time from block needle insertion to the time of completion of local anesthetic injection (min) |
| | Needle passes | Number of times the needle was re-advanced after being withdrawn |
| | Block duration | Time from block completion to the patient reported time of noticeable increase in pain (min) |
| | Block failure | Lack of sensory loss or the need for a repeat block in recovery room |
| | Perioperative opioid consumption | Sum of opioid administered (preoperative, intraoperative, recovery room and oral opioid taken by patient at home in the 1st 24 hours after the block completion) in oral morphine equivalents (mg) |
| | Pain scores | Assessed using the Numerical Rating Scale (NRS) from 0–10, where zero is no pain and 10 is the worst imaginable pain |
| | Recovery room length of stay | Difference in time from arrival to the recovery room until discharge criteria were met |
| | PONV | Need for any antiemetic in the recovery room |
| | Patient satisfaction | Rated by patients (0–10) with zero meaning unsatisfied, five meaning neutral, and ten meaning extremely satisfied |
| **Safety** | Paresthesia | Abnormal sensation of "tingling, pins and needles or electric shock radiating down the arm" during block placement |
| | Horner's syndrome | Ptosis and miosis of the ipsilateral pupil |
| | Dyspnea | Shortness of breath as described by patient |
| | Hoarseness | Voice changes as described by patient |

Total opioid consumption was calculated as total oral morphine equivalents, using a previously published equation [9,10] which is included in S2 File.

**Safety outcomes.** The secondary outcomes relating to safety of the block included the incidence of paresthesia, Horner's syndrome, dyspnea, hoarseness, as described in Table 1, as well as nerve injury, local anesthetic toxicity and infection at site of block injection. If patients reported any residual numbness or weakness during the postoperative day 1 phone call, then they were contacted daily by the study team until it resolved.

## Sample size calculation

The total sample size for a two-sample comparison of survivor functions for the primary outcome variable of time to onset of motor block was calculated to be 104 patients per group (208 patients in total). In accordance with Freedman's method, the number of events was calculated as 191 with a two-sided α level of 5%, power of 80% and a hazard ratio of 0.66. The total sample size was set at 208 patients to account for patients administratively censored. The proportional hazard assumption was assessed using the log-log plot.

## Statistical analysis

Statistical analysis was performed using STATA 13.1 (StataCorp, College Station, TX). Data was analyzed by intention-to-treat. Data were reported as mean (SD) for continuous variables, as median (interquartile ranges) for non-normally distributed continuous variables and as count (percentage) for categorical variables. Time-to-event data was compared using the log-rank test. Categorical data was analyzed using Chi squared analysis. Comparison of means was performed using the Mann-Whitney U test.

## Results

A total of 208 ASA I-II adults were enrolled in this study. Due to simple randomization, there were 105 patients in the extraplexus group and 103 patients in the intraplexus group (Fig 1). After enrollment and allocation, 3 patients in the extraplexus group had a small amount of local anesthetic deposited in the intraplexus space, while 8 in the intraplexus group had a small amount of local anesthetic deposited in the extraplexus space. These patients were analyzed in an intention-to-treat manner. There were two patients (1 in extraplexus group, 1 in intraplexus group) that were missing post-operative sensory exam data and three patients who were lost to telephone follow-up on postoperative day 1; this missing data was excluded from analysis in the post-operative sensory exam analysis and postoperative day 1 analysis, respectively.

The proportional hazard assumption was confirmed for the primary outcome, time to loss of shoulder abduction.

There were no differences in baseline characteristics between the two groups (Table 2). Surgical technique did not vary between surgeons. A similar percentage of patients in both groups converted to open shoulder surgery, which was open biceps tenodesis (Table 2). The patient, operating room team and post-anesthesia care team were blinded to group assignment throughout the study. However, in approximately 10% of the enrolled patients, the outcome assessor performing the sensory and motor assessments was aware of the study group assignment.

### Efficacy outcomes

Loss of shoulder abduction (motor block) was faster in the intraplexus group (log-rank p-value <0.0005; Fig 2). The median value of time to onset of motor blockade was 4 minutes

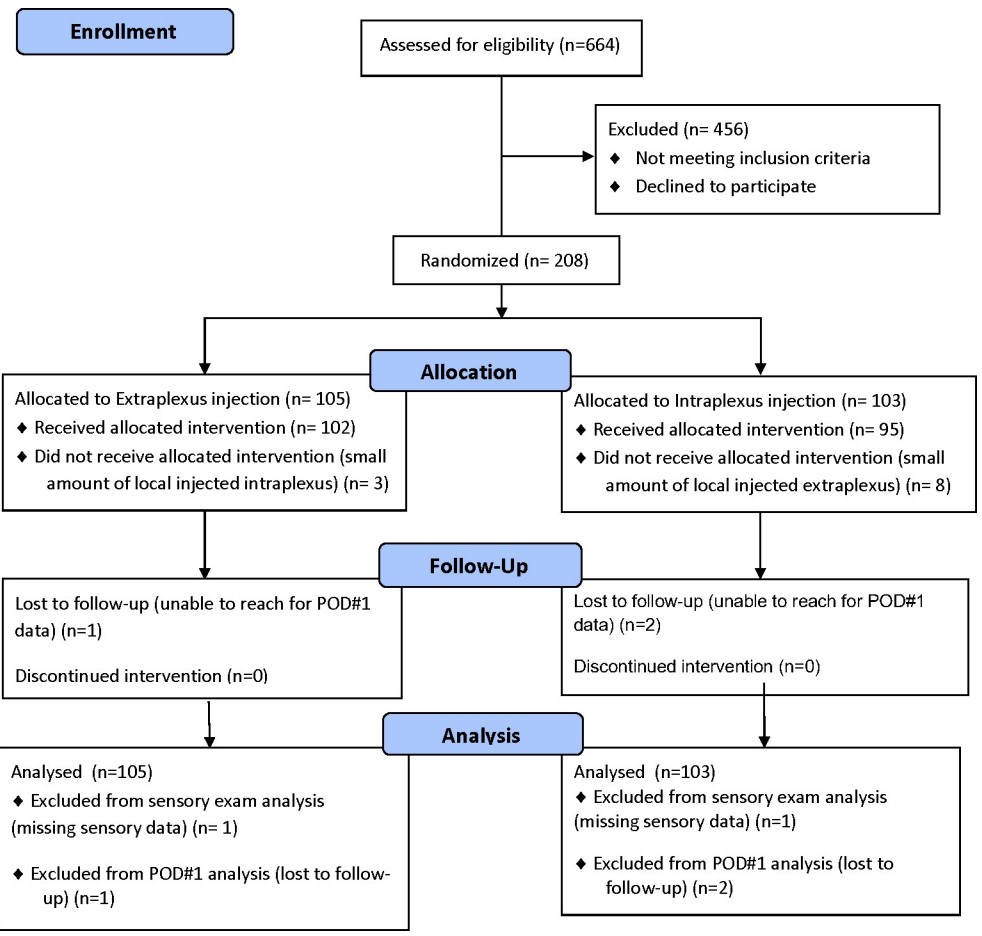

**Fig 1. CONSORT diagram of patient recruitment.**

[interquartile range: 2–6] in the intraplexus group compared to 6 minutes in the extraplexus group [4–10], p<0.0005). The onset of sensory blockade was faster in the intraplexus group for the axillary nerve (p = 0.0013), musculocutaneous nerve (p = 0.0001), median nerve (p = 0.0128) and radial nerve (p = 0.0046) distributions (Fig 3). No difference was noted in the

**Table 2. Patient demographics and baseline characteristics.**

|  | Extraplexus n = 105 | Intraplexus n = 103 |
|---|---|---|
| **Age; y** | 51.0 (14.2) | 47.1 (16.7) |
| **Height; cm** | 173.7 (8.9) | 174.4 (9.3) |
| **Weight; kg** | 78.0 (14.3) | 79.1 (16.2) |
| **BMI; kg/m$^2$** | 25.8 (4.02) | 25.9 (4.28) |
| **Sex; female** | 31 (29.5%) | 30 (29.1%) |
| **Surgeon A** | 69 (65.7%) | 67 (65.0%) |
| **Duration of surgery; min** | 76.4 (25.0) | 71.7 (23.2) |
| **Conversion to open surgery** | 12 (11.4%) | 12 (11.7%) |

Values are number (proportion) or mean (SD).

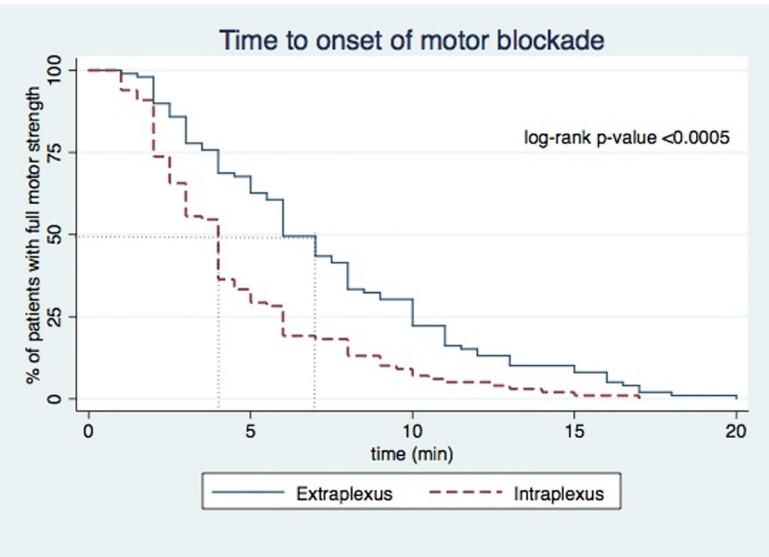

**Fig 2. Kaplan-Meier curve of time to onset of motor blockade.** Extraplexus (solid line); intraplexus (dashed line).

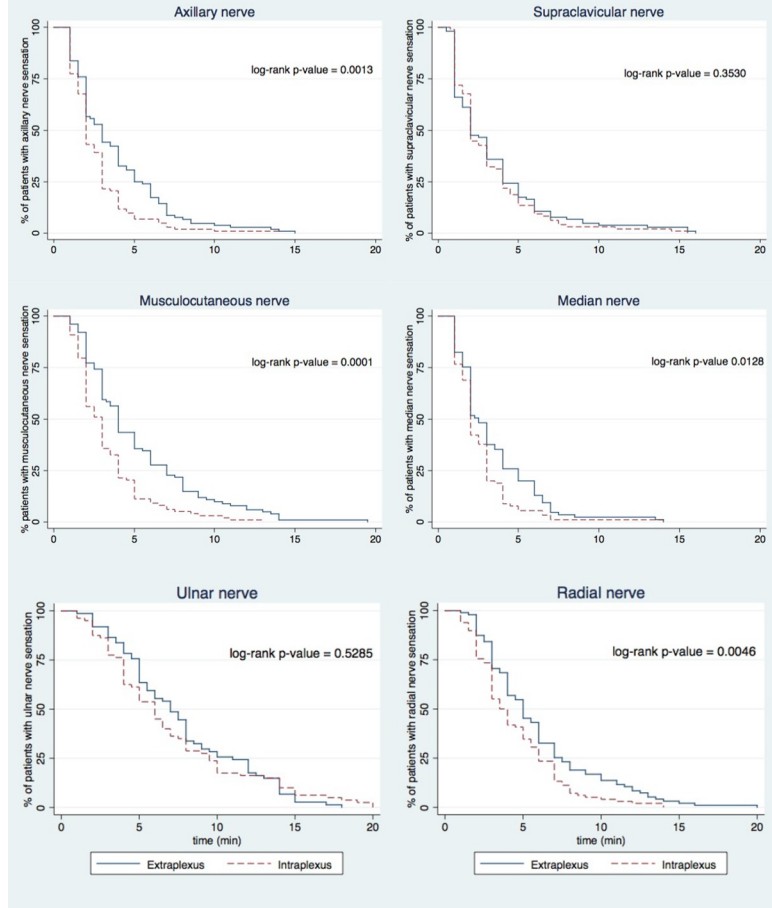

**Fig 3. Kaplan-Meier curve of time to onset of sensory blockade.** Extraplexus (solid line); intraplexus (dashed line).

**Table 3. Block characteristics and efficacy outcomes.**

| | Extraplexus n = 105 | Intraplexus n = 103 | *P*-value |
|---|---|---|---|
| **Block performance time; min** | 4.0 [3.0–5.0] | 4.0 [3.0-.0] | 0.060 |
| **# of needle passes** | 3 [2–3] | 2 [1–3] | p<0.001* |
| **Block duration; min** | 899.6 (267.2) | 961.4 (274.9) | 0.106 |
| **Recovery room length of stay; min** | 84.0 [69.0–99.0] | 82.0 [66.0–103.0] | 0.943 |
| **Satisfaction score (0–10)** | 10 [9–10] | 10 [9–10] | 0.931 |

Values are number (proportion), mean (SD) or median [IQR].

* *P* < 0.05.

onset of sensory blockade for supraclavicular and ulnar nerve distributions between the two groups.

The intraplexus approach required fewer needle passes than the extraplexus approach (2 vs. 3, p<0.0005); however, there was no difference in block performance time between the two groups (Table 3). There was no difference in perioperative opioid consumption, block duration, highest NRS pain score in the recovery room, recovery room length of stay between groups, nor highest NRS pain score at 1 week postoperatively (Tables 3 and 4). There were no failed blocks as confirmed by sensory testing in the recovery room and no patients required a rescue block. Overall, patients were highly satisfied with their care with no difference between the two groups.

## Safety outcomes

The intraplexus approach resulted in a higher incidence of paresthesia during block placement (35.9% vs. 14.5%, p = 0.0004). The longest duration of paresthesia postoperatively was 7 days, which occurred in one patient that was in the intraplexus group. Both groups exhibited equal incidence of Horner's syndrome, dyspnea, hoarseness, and PONV (Table 5). No complications associated with nerve blocks (nerve injury, local anesthesia toxicity, nor infection) were observed in the study.

## Discussion

Although an intraplexus injection of local anesthetic for interscalene block can result in the faster onset of motor and sensory brachial plexus blockade, the difference is not clinically significant. Further in this study, we found no difference in block duration, success rate, or

**Table 4. Pain scores and opioid consumption data.**

| | Extraplexus n = 105 | Intraplexus n = 103 | *P*-value |
|---|---|---|---|
| **Total perioperative opioid consumption; mg oral morphine equivalents (OME)** | 66.3 [50–91.3] | 70 [50–98.8] | 0.537 |
| Preoperative opioid consumption for block placement; mg OME | 0 [0–0] | 0 [0–0] | 0.303 |
| Intraoperative opioid consumption; mg OME | 15 [0–30] | 15 [0–30] | 0.462 |
| Recovery room opioid consumption; mg OME | 0 [0–0] | 0 [0–0] | 0.290 |
| Postoperative day 1 opioid consumption; mg OME | 45.0 [30.0–70.0] | 50.0 [30.0–75.0] | 0.387 |
| **Highest NRS pain score in recovery room†** | 0 [0–1] | 0 [0–1] | 0.993 |
| **Highest NRS pain score at rest on postoperative day 1†** | 3 [2–5.5] | 4 [2–6] | 0.237 |
| **Highest NRS pain score with movement on postoperative day 1†** | 7 [4.5–8.5] | 7 [5–9] | 0.376 |
| **Highest NRS at one week postoperative visit†** | 2 [0–4] | 2 [1–4] | 0.333 |

Values are median [IQR].

†NRS; numerical rating scale (0–10), OME; oral morphine equivalents.

**Table 5. Safety outcomes.**

|  | Extraplexus n = 105 | Intraplexus n = 103 | *P*-value |
|---|---|---|---|
| **Paresthesias** | 15 (14.7%) | 37 (35.9%) | 0.0004* |
| **Horner's syndrome** | 12 (11.4%) | 17 (16.5%) | 0.291 |
| **Dyspnea** | 7 (6.7%) | 8 (7.8%) | 0.774 |
| **Hoarseness** | 2 (1.9%) | 2 (1.9%) | 0.992 |
| **PONV** | 7 (6.9%) | 6 (6%) | 0.803 |

Values are number (proportion).

* *P* < 0.05.

analgesic benefit between intraplexus and extraplexus approaches to interscalene brachial plexus injection. The higher incidence of paresthesia in the intraplexus group compared to the extraplexus group is particularly concerning. Although the rate of post-block neuralgia associated with transient paresthesia is not well defined, there are multiple cases of chronic debilitating pain associated with paresthesia techniques, which most likely reflect an intraneuronal injection of local anesthetic [11]. Fortunately, in our study, any paresthesia associated with the block was transient, with the majority resolving during block placement with adjustment of the needle and there were no long-term neurologic deficits. Notably, the incidence of paresthesia in our study was much lower than the 96.7% incidence seen with intraplexus interscalene injection by Maga and colleagues [7]. Given how similar the block profile is between the two approaches, in terms of onset, duration, and analgesic benefit, we believe the results of this study call into question the benefits and safety of intraplexus injection for interscalene block.

The results of this study are fairly consistent with other studies that have also examined the efficacy of intraplexus and extraplexus injection for interscalene block. As in this study, Spence and colleagues also concluded that there is little advantage to intraplexus injection [6]. Unfortunately, the study conducted by Spence et al was designed as a noninferiority study and the anesthetic protocol was changed mid-enrollment [6]. Ultimately, their study was terminated early due to a difficulty enrolling patients after a clinical practice change unrelated to the safety or efficacy of interscalene blocks. Further, their extraplexus injection only consisted of injection posterior to the brachial plexus in the interscalene groove whereas in our study the injection was both anterior and posterior to the brachial plexus. Of note, there is a potential risk of phrenic nerve injury with an extraplexus approach that traverses above C5 to the space anterior to the brachial plexus and in close proximity to the anterior scalene muscle. However, in our study, there was no difference in the incidence of dyspnea in either group.

Maga and colleagues found a significant difference in onset of complete motor and sensory blockade between intraplexus and extraplexus injection for interscalene nerve block with 70% of the intraplexus approach blocked at 10 minutes compared to only 37% in the extraplexus group [7]. A notable difference between Maga's study and our study is that Maga's extraplexus approach involved out-of-plane intramuscular injections in the scalene muscles adjacent to the brachial plexus and in our study, the extraplexus injection was performed in-plane and not intramuscular, but rather in the space between the scalene muscles and outside of the brachial plexus sheath. This difference in injection approach would likely affect the distribution of the local anesthetic and potentially affect efficacy and safety profiles. Injecting directly into the muscle to perform interscalene brachial plexus blocks is particularly concerning given the growing evidence of the local anesthetic-induced myotoxicity [12].

There is growing evidence that an intraplexus injection for interscalene block can increase the potential for neurologic injury. In a cadaveric study by Szerb and colleagues, 11.5% of

intraplexus injections were actually subepineural, while there were no subepineural injections in the periplexus group [13]. Another concern with intraplexus injection is if providers inject between what may appear as C6 and C7, they are likely to have an intraneural injection, as C6 can frequently have an intraroot splitting in the interscalene groove [14]. Unfortunately, the use of ultrasound does not necessarily protect against intraneural injection as in an observational study of 257 patients receiving ultrasound-guided interscalene and supraclavicular nerve blocks, 17% of patients had an unintentional intraneural injection [15]. Furthermore, in a cadaveric study by Orebaugh et al., there was 50% rate of unintentional injection into the subepineurium of the brachial plexus during ultrasound-guided interscalene nerve block [16]. Permanent neurologic injury has occurred with interscalene block even with the use of ultrasound guidance [17].

This study has some limitations. Although patients and providers administering medications intraoperatively and postoperatively were always blinded to the treatment groups, the staff assessing sensory and motor blockade were not blinded to group assignment in approximately 10% of the patients. While not ideal, we still consider the risk of bias as low [18], as the patients' ability to abduct the shoulder is unlikely to be influenced by the lack of blinding of the outcome assessor. Shoulder abduction is a rather objective outcome, especially when blinding of the patients themselves is intact. In addition, the proportion of cases with broken blinding of the outcome assessor was small (20 of 208 patients). Another limitation was the relatively high volume of local anesthetic used in this study, which was commonly used at the time that this study was designed. One could argue that not as much difference was seen between the groups due to the large volume of local anesthetic use, however, we were able to see a difference in motor onset in this study which strengthens the results of this study. Further, larger amounts of local anesthetics were used in the studies by Spence et al and Maga et al [6,7]. An additional limitation of this study is the reliance on patient report for the duration of the block, which can be a subjective metric. In this study, patient-reported dyspnea was used as a surrogate of hemidiaphragmatic paralysis; however, this is not as precise as diaphragmatic function measured using ultrasound or spirometry and represents another limitation of this study.

In summary, the intraplexus approach to the interscalene brachial plexus block results in a faster motor onset than the extraplexus approach. It also results in a faster sensory block, requires fewer needle passes, and provides equally effective analgesia as the extraplexus approach. However, the time difference in block onset between groups is small and not a clinically meaningful difference. Given that the intraplexus group was associated with increased incidence of transient paresthesia and there were few differences in block quality between intraplexus and extraplexus approaches, an extraplexus approach to interscalene brachial plexus block may be a more appropriate choice.

## Supporting information

**S1 File. Ultrasound images of extraplexus versus intraplexus injection.**
(DOCX)

**S2 File. Oral morphine equivalence conversion equation.**
(DOCX)

**S3 File. Anonymized data set.**
(XLS)

**S4 File. CONSORT checklist.**
(DOC)

**S5 File. Interscalene study protocol.**
(DOC)

## Acknowledgments

The authors would like to thank Dr. Jeffrey Ghassemi for his help in patient recruitment and data collection.

## Author Contributions

**Conceptualization:** Monica W. Harbell, Pedram Aleshi.

**Data curation:** Monica W. Harbell, Sakura Kinjo, Edward Yap, Matthias R. Braehler, Pedram Aleshi.

**Formal analysis:** Monica W. Harbell, Kerstin Kolodzie, Matthias Behrends, C. Benjamin Ma, Sakura Kinjo, Edward Yap, Matthias R. Braehler, Pedram Aleshi.

**Investigation:** Monica W. Harbell, Kerstin Kolodzie, Matthias Behrends, C. Benjamin Ma, Sakura Kinjo, Edward Yap, Matthias R. Braehler, Pedram Aleshi.

**Methodology:** Monica W. Harbell, Kerstin Kolodzie, Matthias Behrends, Sakura Kinjo, Edward Yap, Matthias R. Braehler, Pedram Aleshi.

**Project administration:** Monica W. Harbell.

**Validation:** Monica W. Harbell.

**Writing – original draft:** Monica W. Harbell.

**Writing – review & editing:** Monica W. Harbell, Kerstin Kolodzie, Matthias Behrends, C. Benjamin Ma, Sakura Kinjo, Edward Yap, Matthias R. Braehler, Pedram Aleshi.

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
