## [Decision Letter · Decision Letter 0]

9 Sep 2020

PONE-D-20-12274

Extraplexus versus intraplexus ultrasound-guided interscalene brachial plexus block for ambulatory arthroscopic shoulder surgery: a randomized controlled trial

PLOS ONE

Dear Dr. Harbell,

Thank you for submitting your manuscript to PLOS ONE. After careful consideration, we feel that it has merit but does not fully meet PLOS ONE’s publication criteria as it currently stands. Therefore, we invite you to submit a revised version of the manuscript that addresses the points raised during the review process.

Three experts in the field reviewed your manuscript. They would like further details, clarifications, and discussions of several points.

We look forward to receiving your revised manuscript.

Kind regards,

Susan Hepp

Academic Editor

PLOS ONE

Journal Requirements:

2.We note that you have indicated that data from this study are available upon request. PLOS only allows data to be available upon request if there are legal or ethical restrictions on sharing data publicly. For information on unacceptable data access restrictions, please see http://journals.plos.org/plosone/s/data-availability#loc-unacceptable-data-access-restrictions.

Reviewers' comments:

Reviewer's Responses to Questions

**Comments to the Author**

1. Is the manuscript technically sound, and do the data support the conclusions?

Reviewer #1: Yes

Reviewer #2: Yes

Reviewer #3: Yes

2. Has the statistical analysis been performed appropriately and rigorously? 

Reviewer #1: Yes

Reviewer #2: Yes

Reviewer #3: Yes

3. Have the authors made all data underlying the findings in their manuscript fully available?

Reviewer #1: Yes

Reviewer #2: Yes

Reviewer #3: Yes

4. Is the manuscript presented in an intelligible fashion and written in standard English?

Reviewer #1: Yes

Reviewer #2: Yes

Reviewer #3: Yes

5. Review Comments to the Author

Reviewer #1: This RCT was designed to compare the efficacy and safety of the two injection techniques (intraplexus vs extraplexus) for interscalene brachial plexus block. This work repeats the work of some other studies, as the authors acknowledge in the discussion.

Comments:

1) Introduction

1.1) Page 4 Line 2-5: The references 1-5 are quite outdated. There are several recent well-designed comparative studies concerning benefits of ultrasound-guided interscalene brachial plexus block in arthroscopic shoulder surgery. Please update the references.

1.2) Page 4 Line 5-6: The optimal location of LA injection for interscalene brachial plexus block is not actually “unknown”. It could be concluded from the results of Spence et al. and Maga et al. that the extraplexus injection was suggested because the rate of transient paresthesia was higher during the intraplexus injection, which possibly increased risk of neurologic complications. Please restate the sentence and add these references in the introduction.

2) Methods

2.1) I agree to the authors that a high volume of local anesthetic potentially masks the differences in efficacy of the two techniques. A typical volume of local anesthetic for the interscalene brachial plexus block is 10-20 ml. Therefore, please provide references and reasons why the authors used 25 ml of local anesthetic.

2.2) Because the main intraoperative anesthetic technique in this study was general anesthesia, the onset times of motor and sensory blockade are not absolutely important. Please provide a rationale of choosing time to loss of shoulder abduction as a primary outcome in the manuscript.

2.3) Some of the patients coming for arthroscopic shoulder surgery have preoperative limit of the shoulder joint motion and could not abduct the shoulder to 90°. Please explain how to assess the primary outcome in this group of patients and how is the reliability of these data.

2.4) Please clarify the definitions of sensory and motor blockade whether they are complete or decreased loss of sensation and motor function.

2.5) The supraclavicular nerve is not a part of brachial plexus. Please provide a reason of evaluating a sensory blockade of this nerve after the interscalene brachial plexus block and please explain how this nerve, which originates from the C3-4 nerve roots, is anesthetized by this procedure.

2.6) Because this study is a prospective study, postoperative pain management protocol should initially be established. Dosage and times of oral analgesics including acetaminophen and NSAIDs should be uniformed, while type, dosage and indication of rescue opioids should be clearly determined. Therefore, pain scores and opioid consumption data in this study have to be interpreted with caution.

2.7) Because transient paresthesia is expected in the intraplexus group, please provide an explanation of the instruction when paresthesia occurred.

3) Results

3.1) Please provide any references that support or advice against an incomplete blinding of the outcome assessors and discuss about risk of bias in the manuscript.

3.2) Please discuss why there was no difference in the onset of sensory blockade of ulnar and supraclavicular nerve between the two groups. From Figure 4, in both groups, there were more than 90% of patients with sensory loss in the ulnar and supraclavicular area at 15-20 minutes after the block.

3.3) In the safety outcomes, all data recorded are complications of the interscalene brachial plexus block except postoperative nausea and vomiting which is a complication of opioid or anesthetics. If the authors would like to show the safety outcomes of the block, nausea and vomiting could be discarded.

4) Discussion

One major safety concern of the interscalene brachial plexus block is hemidiaphragmatic paralysis. Evaluation of dyspneic symptom after the block is not sensitive enough to detect this condition. Please consider adding in the limitation regarding a lack of diaphragm assessment after the procedure.

Reviewer #2: Thank you for this work. This is a RCT comparing extraplexus with intraplexus drug deposition for interscalene block for arthroscopic shoulder study.

The study is well done and the manuscript is clear and understandable. The volume used for the block is excessive and in the present day, no one uses that much. However, the authors have addressed this.

Though the study has a few negative results, it does add to the knowledge of interscalene block technique.

One of a major issue is the block wearing off time. It is vague, at best. Once the patients starts having pain, authors considered it as the block end time. So if the pain increased from NRS 2 to NRS 4, would that be considered as block wearing off? Or did the authors wait until it became 8/10? Was the patient asked to record the time this happened?

There are a few minor points that I wish to ask the authors:

1. What is meant by surgeon A in table 1?

2. Units for opioid consumption in table 3: should be mentioned for each row

3. In the extraplexus approach: when the needle goes above C5, there may be a chance of getting the phrenic nerve that lives in that little area above the C5, between the sternocleidomastoid and the anterior scalene muscle. This should be addressed.

4. There are many definitions in the 'efficacy outcomes' section. I suggest that all definitions be in a table so that all can be seen at once. This also occupies less space.

5. The same goes for the opioid conversion formula. Is it really necessary? It can either be added as a table or as an appendix if it really must be included. For conversion, it would have been better if an internationally known conversion table was used.

6. NRS should be explained in its entire form when first used. It is explained later in the manuscript but used earlier.

7. PONV does not need to be explained every time it is used. Full form explained at the time of first use is enough.

8. Page 14, line 7. How and why was the supraclavicular nerve block tested? As the authors may know, supraclavicular nerve is different from the supraclavicular approach to the brachial plexus. Actually, supraclavicular nerve is not part of the brachial plexus at all. It arises from the superficial cervical plexus and is derived from C3-4 nerve roots.

Reviewer #3: This is a well-described clinical trial comparing time to loss of abduction for extra vs intra plexus injection. The randomization procedure was specified, the CONSORT guidelines were followed, and the analysis plan standard for time-to-event studies. The sample size computation was based on "Freedman's method", which is not referenced. I am not familiar with this. What are the assumptions (exponential distribution, censoring?) Were these assumptions met in the actual trial? This should be mentioned in the discussion. Was there a clinically relevant reason for a 1/3 reduction in hazard?

6. PLOS authors have the option to publish the peer review history of their article (what does this mean?). If published, this will include your full peer review and any attached files.

Reviewer #1: No

Reviewer #2: **Yes: **Shalini Dhir

Reviewer #3: No

---

## [Author Response · Author response to Decision Letter 0]

20 Oct 2020

Journal Requirements:

• Done

2.We note that you have indicated that data from this study are available upon request. PLOS only allows data to be available upon request if there are legal or ethical restrictions on sharing data publicly. For information on unacceptable data access restrictions, please see http://journals.plos.org/plosone/s/data-availability#loc-unacceptable-data-access-restrictions.

• The data set will be included as a supporting information file.

Review Comments to the Author

Reviewer #1: This RCT was designed to compare the efficacy and safety of the two injection techniques (intraplexus vs extraplexus) for interscalene brachial plexus block. This work repeats the work of some other studies, as the authors acknowledge in the discussion.

Comments:

1) Introduction

1.1) Page 4 Line 2-5: The references 1-5 are quite outdated. There are several recent well-designed comparative studies concerning benefits of ultrasound-guided interscalene brachial plexus block in arthroscopic shoulder surgery. Please update the references.

• Thank you for the suggestion. We have updated the original references 1-5 to include the following more recent references: 

o Hadzic A, Williams BA, Karaca PE, et al. For outpatient rotator cuff surgery, nerve block anesthesia provides superior same-day recovery after general anesthesia. Anesthesiology 2005;102:1001–7.

o Warrender WJ, Syed UAM, Hammoud S, Emper W, Ciccotti MG, Abboud JA, et al. Pain management after outpatient shoulder arthroscopy: a systematic review of randomized controlled trials. Am J Sports Med 2017; 45: 1676-86.

o Fontana C, Di Donato A, Di Giacomo G, Costantini A, De Vita A, Lancia F, Caricati A. Postoperative analgesia for arthroscopic shoulder surgery: a prospective randomized controlled study of intraarticular, subacromial injection, interscalene brachial plexus block and intraarticular plus subacromial injection efficacy. Eur J Anaesthesiol. 2009; 26:689–693.

1.2) Page 4 Line 5-6: The optimal location of LA injection for interscalene brachial plexus block is not actually “unknown”. It could be concluded from the results of Spence et al. and Maga et al. that the extraplexus injection was suggested because the rate of transient paresthesia was higher during the intraplexus injection, which possibly increased risk of neurologic complications. Please restate the sentence and add these references in the introduction.

• We have added these references to the Introduction and have rephrased the sentence (page 4).

“Previous studies that compared extraplexus to intraplexus injections have had conflicting results with one reporting no difference in block onset (6) and the other reporting a faster onset with the intraplexus approach, but increased paresthesia (7).”

2) Methods

2.1) I agree to the authors that a high volume of local anesthetic potentially masks the differences in efficacy of the two techniques. A typical volume of local anesthetic for the interscalene brachial plexus block is 10-20 ml. Therefore, please provide references and reasons why the authors used 25 ml of local anesthetic.

• We appreciate Reviewer#1’s comment. In our search of the literature, we found a large range of volumes used for the interscalene block. According to the usra.ca website, the usual volume of local anesthetic administration for the interscalene block ranges from 15mL to 40mL. At time that our study was designed, higher volumes were commonly used for interscalene block, in hope that a larger dose would lead to prolonged analgesia. In the interscalene studies performed by Spence and Maga, 30mL were used as the volume of local anesthetic. 

References:

• Usra.ca/regional-anesthesia/specific-blocks/upper-limb/interscaleneblock.php

• Maga J, Missair A, Visan A, Kaplan L, Gutierrez JF, Jain AR, et al. Comparison of Outside Versus Inside Brachial Plexus Sheath Injection for Ultrasound-Guided Interscalene Nerve Blocks. J Ultrasound Med [Internet]. 2016 Feb;35(2):279–85. 

• Spence BC, Beach ML, Gallagher JD, Sites BD. Ultrasound-guided interscalene blocks: understanding where to inject the local anaesthetic. Anaesthesia [Internet]. 2011 Jun; 66(6):509–14. 

2.2) Because the main intraoperative anesthetic technique in this study was general anesthesia, the onset times of motor and sensory blockade are not absolutely important. Please provide a rationale of choosing time to loss of shoulder abduction as a primary outcome in the manuscript.

• While we used general anesthesia in this study, we were interested in making the study results generalizable for providers using regional anesthesia alone. Loss of shoulder abduction is easy to define and measure and reliably indicates the achievement of a dense block. Of note, we chose the same primary outcome as Spence et al. which at the time of study design, was the only published study on this topic.

2.3) Some of the patients coming for arthroscopic shoulder surgery have preoperative limit of the shoulder joint motion and could not abduct the shoulder to 90°. Please explain how to assess the primary outcome in this group of patients and how is the reliability of these data.

• We appreciate this comment from Reviewer #1. Each patient was assessed preoperatively for the ability to abduct the shoulder to 90 degrees. If they were unable to abduct the shoulder to 90 degrees, they were excluded from the study. This was clarified in the Methods section (Page 5). 

2.4) Please clarify the definitions of sensory and motor blockade whether they are complete or decreased loss of sensation and motor function.

• Motor blockade was defined as complete loss of shoulder abduction, while sensory blockade was defined as time to decreased sensation compared to the contralateral side. This was clarified in the Methods section of the manuscript. 

“Patients were assessed for time to complete loss of shoulder abduction by having the patient sit upright and attempt to abduct the arm to 90 degrees. Sensation was assessed using pinprick and comparing to the contralateral side in the sensory distributions of the brachial plexus (axillary, supraclavicular, musculocutaneous, median, ulnar and radial). The time when the sensation first started to feel less sharp compared to the contralateral, non-blocked side was recorded to the nearest minute.”

2.5) The supraclavicular nerve is not a part of brachial plexus. Please provide a reason of evaluating a sensory blockade of this nerve after the interscalene brachial plexus block and please explain how this nerve, which originates from the C3-4 nerve roots, is anesthetized by this procedure.

• We appreciate Reviewer #1’s comment and agree that the supraclavicular nerve is not part of the brachial plexus. Given that the supraclavicular nerve is often blocked by an interscalene block and this blockade contributes to analgesia after shoulder surgery, we have opted to include the evaluation of the sensory blockade of this nerve. 

2.6) Because this study is a prospective study, postoperative pain management protocol should initially be established. Dosage and times of oral analgesics including acetaminophen and NSAIDs should be uniformed, while type, dosage and indication of rescue opioids should be clearly determined. Therefore, pain scores and opioid consumption data in this study have to be interpreted with caution.

• We thank Reviewer #1 for this comment. While we realize the advantages of a uniform postoperative pain management regimen, all oral analgesics in this study were given post-discharge as PRN medications. The oral analgesics were prescribed based on patient preferences and tolerances. We realize that this approach is less precise and respect Reviewer #1’s comment, however, we think that this is a more pragmatic, patient-centered outcome and is more relevant for clinical care given most arthroscopic shoulder surgeries are performed on an ambulatory basis with patients having a say in their analgesic regimen. 

2.7) Because transient paresthesia is expected in the intraplexus group, please provide an explanation of the instruction when paresthesia occurred.

• Thank you for the opportunity to clarify this aspect of the study. The patients were instructed to notify the providers with any paresthesia (pin and needles sensation, sensation of electricity going down the arm) during the block performance. The anesthesia providers who performed the block were instructed to stop any injection with any patient-reported paresthesia, redirect the needle and to not resume any injection until the paresthesia had resolved. We have added these clarifying details to the Methods section of the manuscript (Page 7). 

“Patients were instructed to notify the study team if they experienced any paresthesia during block placement. Anesthesia providers were instructed to stop the block procedure if a paresthesia occurred, to redirect the needle, and to not resume local anesthetic administration until the paresthesia resolved.”

3) Results

3.1) Please provide any references that support or advice against an incomplete blinding of the outcome assessors and discuss about risk of bias in the manuscript.

While all patients as well as the operating room team and post-anesthesia care team were blinded to group assignment throughout the study, in approximately 10% of the enrolled patients (20 out of 208), the outcome assessor performing the sensory and motor assessments was aware of the study group assignment due to staffing limitations.

Even though we cannot completely exclude detection bias due to knowledge of the allocated interventions by the assessor especially of the primary outcome (time to loss of shoulder abduction), we consider the risk of bias as low1: The patients’ ability to abduct the shoulder is unlikely to be influenced by the lack of blinding of the outcome assessor. Shoulder abduction is a rather objective outcome, especially when blinding of the patients themselves is intact. Even suggestive questioning or generous interpretation of arm abduction ability of the outcome assessor is unlikely to influence the primary outcome of time to loss of shoulder abduction in a relevant order. This is especially true considering the small proportion of cases with broken blinding of the outcome assessor (about 20 out of 208 patients). 

We have revised the corresponding paragraph in the result’s section and amended the discussion accordingly.

1. Higgins JPT, Savović J, Page MJ, Elbers RG, Sterne JAC. Chapter 8: Assessing risk of bias in a randomized trial. In: Higgins JPT, Thomas J, Chandler J, Cumpston M, Li T, Page MJ, Welch VA (editors). Cochrane Handbook for Systematic Reviews of Interventions version 6.1 (updated September 2020). Cochrane, 2020. Chapter 8.6., section 5: Whether the assessment of outcome is likely to be influenced by knowledge of intervention received. Available from https://training.cochrane.org/handbook/current/chapter-08#section-8-6

3.2) Please discuss why there was no difference in the onset of sensory blockade of ulnar and supraclavicular nerve between the two groups. From Figure 4, in both groups, there were more than 90% of patients with sensory loss in the ulnar and supraclavicular area at 15-20 minutes after the block.

• We thank the reviewer for bringing up these results from our paper. The interscalene block is performed at C5-6, however often supraclavicular (C3,4) and ulnar nerves (C8, T1) are blocked likely due to overflow of local anesthetic to these nerves. As the blockade of these more distal areas of the brachial plexus relies on overflow spread of local anesthesia, this may explain why there was no difference in onset of sensory blockade for the supraclavicular and ulnar nerves.

3.3) In the safety outcomes, all data recorded are complications of the interscalene brachial plexus block except postoperative nausea and vomiting which is a complication of opioid or anesthetics. If the authors would like to show the safety outcomes of the block, nausea and vomiting could be discarded.

• We agree with the reviewer that postoperative nausea and vomiting (PONV) is associated with opioids or anesthetics, however, a better nerve block or prolonged analgesia can result in less opioid consumption and less opioid related side effects, including PONV. Thus, we believe that PONV is a valid outcome to report in the manuscript.

4) Discussion

One major safety concern of the interscalene brachial plexus block is hemidiaphragmatic paralysis. Evaluation of dyspneic symptom after the block is not sensitive enough to detect this condition. Please consider adding in the limitation regarding a lack of diaphragm assessment after the procedure.

• We agree that a limitation of this study was that the diaphragmatic function was not directly assessed after the block performance. Of note, when this study was designed, the use of ultrasound to measure diaphragmatic function was not well-described or widely utilized. We have added this limitation to the Discussion as below: 

“In this study, patient-reported dyspnea was used as a surrogate of hemidiaphragmatic paralysis; however, this is not as precise as diaphragmatic function measured using ultrasound or spirometry and represents another limitation of this study.”

Reviewer #2: Thank you for this work. This is a RCT comparing extraplexus with intraplexus drug deposition for interscalene block for arthroscopic shoulder study.

The study is well done and the manuscript is clear and understandable. The volume used for the block is excessive and in the present day, no one uses that much. However, the authors have addressed this.

Though the study has a few negative results, it does add to the knowledge of interscalene block technique.

One of a major issue is the block wearing off time. It is vague, at best. Once the patients starts having pain, authors considered it as the block end time. So if the pain increased from NRS 2 to NRS 4, would that be considered as block wearing off? Or did the authors wait until it became 8/10? Was the patient asked to record the time this happened?

• We agree that block wearing off time can be vague, which is why it was not chosen as a primary outcome. However, we do think of it as a patient-centered outcome, which is both relevant and pragmatic. We asked the patients to note the time that they felt that the block was no longer helping with their pain and record it in a pain diary that was provided to them on the day of the study. Patients can typically notice a difference when the block wears off with a significant increase in pain.

There are a few minor points that I wish to ask the authors:

1. What is meant by surgeon A in table 1?

• There were two surgeons whose cases were included in the study, represented as Surgeon A and B. Table 1 shows the percentage of patients that were operated on by Surgeon A.

2. Units for opioid consumption in table 3: should be mentioned for each row

• We apologize for this oversight and have added these units for each row.

3. In the extraplexus approach: when the needle goes above C5, there may be a chance of getting the phrenic nerve that lives in that little area above the C5, between the sternocleidomastoid and the anterior scalene muscle. This should be addressed.

• We appreciate this comment and have added this potential risk of phrenic nerve injury with the extraplexus approach to the Discussion (Page 18):

“Of note, there is a potential risk of phrenic nerve injury with an extraplexus approach that traverses above C5 to the space anterior to the brachial plexus and in close proximity to the anterior scalene muscle. However, in our study, there was no difference in the incidence of dyspnea in either group.”

4. There are many definitions in the 'efficacy outcomes' section. I suggest that all definitions be in a table so that all can be seen at once. This also occupies less space.

• Thank you for this suggestion. We have added a Table with all of the outcome definitions to improve readability and clarity (new Table 1).

5. The same goes for the opioid conversion formula. Is it really necessary? It can either be added as a table or as an appendix if it really must be included. For conversion, it would have been better if an internationally known conversion table was used.

• We appreciate this comment. In lieu of including the full equation in the manuscript, we will reference a previously published article that utilized the same opioid conversion formula, as well as the original source of the conversion.

6. NRS should be explained in its entire form when first used. It is explained later in the manuscript but used earlier.

• We appreciate this comment and have added the NRS abbreviation explanation earlier in the manuscript when it is first used.

7. PONV does not need to be explained every time it is used. Full form explained at the time of first use is enough.

• Thank you for this suggestion. The abbreviation for PONV is used after it is first use in the manuscript.

8. Page 14, line 7. How and why was the supraclavicular nerve block tested? As the authors may know, supraclavicular nerve is different from the supraclavicular approach to the brachial plexus. Actually, supraclavicular nerve is not part of the brachial plexus at all. It arises from the superficial cervical plexus and is derived from C3-4 nerve roots.

• We appreciate Reviewer #2’s comment and agree that the supraclavicular nerve is not part of the brachial plexus. Given that the supraclavicular nerve is often blocked by an interscalene block and this contributes to the analgesia after shoulder surgery, we have opted to include the evaluation of the sensory blockade of this nerve in this manuscript. 

Reviewer #3: This is a well-described clinical trial comparing time to loss of abduction for extra vs intra plexus injection. The randomization procedure was specified, the CONSORT guidelines were followed, and the analysis plan standard for time-to-event studies. The sample size computation was based on "Freedman's method", which is not referenced. I am not familiar with this. What are the assumptions (exponential distribution, censoring?) Were these assumptions met in the actual trial? This should be mentioned in the discussion. Was there a clinically relevant reason for a 1/3 reduction in hazard?

• The Freedman methods is widely used to calculate sample size when two groups with time to event outcomes are compared using the logrank test. The number of events per group is calculated based on prespecified significance level, power and effect size (expressed as hazard ratio). In a second step, estimated censoring is taken into account. The logrank test assumes proportional hazards. In accordance with Freedman’s method, the number of events was calculated as 191 with a two-sided α level of 5%, power of 80% and a hazard ratio of 0.66. The total sample size was set at 208 patients to account for patients censored. The proportional hazard assumption was assessed and confirmed using the log-log plot. We deemed a one third difference in number of events between study groups at any point of the study period as clinically significant.

Freedman, L. S. 1982. Tables of the number of patients required in clinical trials using the logrank test. Statistics in Medicine 1: 121–129.

---

## [Decision Letter · Decision Letter 1]

30 Nov 2020

PONE-D-20-12274R1

Extraplexus versus intraplexus ultrasound-guided interscalene brachial plexus block for ambulatory arthroscopic shoulder surgery: a randomized controlled trial

PLOS ONE

Dear Dr. Harbell,

Thank you for submitting your manuscript to PLOS ONE. After careful consideration, we feel that it has merit but does not fully meet PLOS ONE’s publication criteria as it currently stands. Therefore, we invite you to submit a revised version of the manuscript that addresses the points raised during the review process.

Please be aware that even though reviewers judged positively, answers to referee #2 are mandatory and must be fully addressed. Please refer to the attached word document.

We look forward to receiving your revised manuscript.

Kind regards,

Johannes Fleckenstein

Academic Editor

PLOS ONE

Reviewers' comments:

Reviewer's Responses to Questions

**Comments to the Author**

1. If the authors have adequately addressed your comments raised in a previous round of review and you feel that this manuscript is now acceptable for publication, you may indicate that here to bypass the “Comments to the Author” section, enter your conflict of interest statement in the “Confidential to Editor” section, and submit your "Accept" recommendation.

Reviewer #1: All comments have been addressed

Reviewer #2: All comments have been addressed

Reviewer #3: All comments have been addressed

2. Is the manuscript technically sound, and do the data support the conclusions?

Reviewer #1: Yes

Reviewer #2: Yes

Reviewer #3: (No Response)

3. Has the statistical analysis been performed appropriately and rigorously? 

Reviewer #1: Yes

Reviewer #2: I Don't Know

Reviewer #3: (No Response)

4. Have the authors made all data underlying the findings in their manuscript fully available?

Reviewer #1: Yes

Reviewer #2: Yes

Reviewer #3: (No Response)

5. Is the manuscript presented in an intelligible fashion and written in standard English?

Reviewer #1: Yes

Reviewer #2: Yes

Reviewer #3: (No Response)

6. Review Comments to the Author

Reviewer #1: The authors have provided rational explanation for all comments and have revised the manuscript as per suggestions.

Reviewer #2: Please see the attached word document for the review.

In question 1 of this, it states that all comments have been addressed. They have been NOT but I cannot submit my review unless I say that. Please note that my comments have not been addressed.

Reviewer #3: (No Response)

7. PLOS authors have the option to publish the peer review history of their article (what does this mean?). If published, this will include your full peer review and any attached files.

Reviewer #1: No

Reviewer #2: **Yes: **Shalini Dhir

Reviewer #3: No

---

## [Author Response · Author response to Decision Letter 1]

11 Jan 2021

January 11, 2021

Re: PONE-D-20-12274 “Extraplexus versus intraplexus ultrasound-guided interscalene brachial plexus block for ambulatory arthroscopic shoulder surgery: a randomized controlled trial”

Dear Editor Johannes Fleckenstein,

Thank you for the opportunity to revise our manuscript PONE-D-20-12274 titled, “Extraplexus versus intraplexus ultrasound-guided interscalene brachial plexus block for ambulatory arthroscopic shoulder surgery: a randomized controlled trial.” We appreciate the reviewers’ comments and have incorporated their feedback. We have addressed each comment below. We believe that this has greatly improved the manuscript and thank you for considering our revised manuscript.

Sincerely,

Monica W. Harbell, MD

Reviewer #2 comments:

1. This is an old study. As the authors mention, it was registered in 2012 and the results were presented in 2016. 

• While a few studies have investigated intraplexus vs. extraplexus interscalene blocks in the past, we believe that this current study adds to the body of literature by providing a more detailed depiction of motor and sensory loss after interscalene block. Further, previous studies have not looked into the safety outcomes of these two approaches.

2. The authors have used a large volume of local anesthetic for the block. They do give a valid reason. 

• We thank the reviewer for understanding why the volume used in this study was chosen. 

3. Sample size 

a. Sample size was not adjusted for dropouts. This always leads to problems when there is missing data, postoperatively. In this study, the sample size for the primary objective was 104 per group. In statistical sense, authors did not reach the sample size that they were supposed to. They only had 103 patients in one group and though the difference is only 1, mathematically and statistically, they did not attain the calculated sample. This problem would not have happened if they had increased their sample by 10% (or any number, for that matter). 

• Sample size calculations for survival data (Freedman method) differ from the usual sample size calculation. It is a composite of two steps: 1. Calculation of the number of events needed to show a statistically significant difference (191 events = number patients with loss of shoulder abduction during our study period). 2. Adjustment of the sample size for administrative censoring. Those are patients that never have an event (sample size increase of 17 to account for possible patients with no loss of shoulder abduction during our study period).

• In our study, it turned out that all of our patients in the study lost shoulder abduction. The number of events for the primary outcome was therefore 208, far exceeding the required number of events of 191. It also turned out that we did not have any dropouts for the primary outcome. All 208 patients were included in the analysis of time to shoulder abduction. In a statistical and mathematical sense, the sample size in our study is sufficient.

b. The authors should explain why there was unequal distribution of patients across the 2 groups resulting in inadequate sample in the intraplexus group (105 vs 103). 

• This is the effect of the simple randomization scheme we chose for our study. This is a well-accepted type of randomization in the scientific community. The uneven numbers between groups are likely to occur with this randomization scheme and are only problematic with small sample sizes (less than 20). In a statistical and mathematical sense, the addition or subtraction of one patient in one group or the other will not affect the results in a trial with a sample size of over 200 participants.

c. The primary outcome was time to block and the sample size was calculated for this outcome only. Paresthesia is the only secondary outcome significant finding (other than the primary outcome). However, it was a secondary outcome and the sample was not calculated for this outcome. This should be mentioned in the limitations.

• It is common in clinical research that the sample size is only calculated for the primary outcome. The reviewer is correct that we cannot conclude that the non-significant secondary outcomes indicate no difference between study groups. However, when there is a significant statistical difference in a secondary outcome, it is not expected that the difference will go away with increasing sample size. It is therefore legitimate to conclude that there was a significant difference in paresthesia between study groups as we have done in this study.

4. In the extraplexus group, one patient was lost to followup but the n in the tables remains 105. Similarly, there were 2 patients in the intraplexus group, but the n remains 103. 

• In the extraplexus group, one patient was excluded from part of the sensory exam analysis as they were missing sensory data for that particular nerve distribution. This patient was included in the rest of the analysis (motor block, POD#1 questions, etc). A different patient in the extraplexus group who was unable to be reached for the POD#1 call was excluded for the POD#1 questions, but all the remaining data was able to be analyzed for that group. Similarly, in the intraplexus group, it was not the same patient was missing both sensory and POD#1 data. The tables contain a mix of data from the day of surgery and POD#1. For this reason, the n remains 105 in the extraplexus group and 103 in the intraplexus group in the tables.

5. Unblinding is a very big limitation in a double-blind trial, however small the percentage may be. It would be best to use single blind and remove the sections explaining 10% blinding and lack of staffing. Lack of staffing is a local matter that everyone faces but never acknowledges in scientific studies. For most ethics review boards, one has to ascertain that they have enough staff, resources and funding to be able to do the study. Therefore, one cannot (should not) say that during the study, they did not have enough staff.

• We appreciate the Reviewer’s comments and revised the manuscript to remove mention of lack of staffing. Given that only a small proportion of the participants experienced unblinding for sensory and motor assessments, we believe that the most accurate way to describe the study is as a double blinded study and we believe that ethically we should disclose that 10% were unblinded.

6. Paresthesia is an important secondary outcome of this study. The authors should spend a few lines describing what they considered paresthesia in the context of this work. 

• We thank the reviewer for this suggestion. We have added a few lines describing what was defined as paresthesia to the Methods section.

7. Conclusions: The conclusions focus on the secondary outcome paresthesia (for which the sample size was not calculated) with no mention of the main purpose of the study (for which the sample size was calculated). This should be rewritten. 

• We have emphasized the primary outcome of this study in the revised version of the Conclusion.

8. References:

a. Some have DOI while some do not. Please be consistent.

b. Ref# 2, 3 and 17 have no year of publication. The authors have mentioned n.d. that I presume is ‘no date’. This is incorrect. Please go on pubmed to find the correct year. 

c. Ref#18 has a special symbol the purpose of which I cannot understand. 

• Our most recent version of the manuscript has 18 references. We have updated and corrected the references with the suggested changes.

---

## [Editor Report · Decision Letter 2]

27 Jan 2021

Extraplexus versus intraplexus ultrasound-guided interscalene brachial plexus block for ambulatory arthroscopic shoulder surgery: a randomized controlled trial

PONE-D-20-12274R2

Dear Dr. Harbell,

We’re pleased to inform you that your manuscript has been judged scientifically suitable for publication and will be formally accepted for publication once it meets all outstanding technical requirements.

Kind regards,

Johannes Fleckenstein

Academic Editor

PLOS ONE
---

## [Editor Report · Acceptance letter]

4 Feb 2021

PONE-D-20-12274R2 

Extraplexus versus intraplexus ultrasound-guided interscalene brachial plexus block for ambulatory arthroscopic shoulder surgery: a randomized controlled trial 

Dear Dr. Harbell:

I'm pleased to inform you that your manuscript has been deemed suitable for publication in PLOS ONE. Congratulations! Your manuscript is now with our production department. 

Kind regards, 

on behalf of

Priv.-Doz. Dr. Johannes Fleckenstein 

Academic Editor

PLOS ONE